# Do Current Multi-Task Optimization Methods in Deep Learning Even Help?

**Derrick Xin**\*
Google Research
Mountain View, CA
dxin@google.com

**Behrooz Ghorbani**\*
Google Research
Mountain View, CA
ghorbani@google.com

**Ankush Garg**
Google Research
Mountain View, CA
ankugarg@google.com

**Orhan Firat**
Google Research
Mountain View, CA
orhanf@google.com

**Justin Gilmer**
Google Research
Mountain View, CA
gilmer@google.com

## Abstract

Recent research has proposed a series of specialized optimization algorithms for deep multi-task models. It is often claimed that these multi-task optimization (MTO) methods yield solutions that are superior to the ones found by simply optimizing a weighted average of the task losses. In this paper, we perform large-scale experiments on a variety of language and vision tasks to examine the empirical validity of these claims. We show that, despite the added design and computational complexity of these algorithms, MTO methods do not yield any performance improvements beyond what is achievable via traditional optimization approaches. We highlight alternative strategies that consistently yield improvements to the performance profile and point out common training pitfalls that might cause suboptimal results. Finally, we outline challenges in reliably evaluating the performance of MTO algorithms and discuss potential solutions.

## 1 Introduction

Multitask models are ubiquitous in deep learning [1, 2, 17]. This popularity stems from the fact that these models can potentially leverage transfer learning in between different tasks and modalities. Moreover, by reducing the number of the models that need to be maintained, multitask models greatly simplify serving users.

Multitask models come with their own challenges and downsides. Different tasks often compete with each other for model capacity, leading to *the task interference* problem. Finding the training setting that strikes the right balance between different tasks is an engineering intensive endeavor that requires extensive trial and error for most realistic setups.

Over the past few years, a vast number of multi-task optimization (MTO) algorithms have been proposed in the literature that claim to alleviate the task interference problem [5, 9, 20, 21, 26, 30, 31]. These algorithms typically leverage clever intuitions about the training process to dynamically balance the different tasks throughout training. However, in exchange for this, these algorithms often drastically add to the computational and design complexity of the training process.

The goal of this paper is not to provide another MTO algorithm. Instead, we provide a large-scale empirical study of the algorithms presented in the literature; we examine to what degree the

---

\*Equal contribution

36th Conference on Neural Information Processing Systems (NeurIPS 2022).

improvements presented in the literature are reproducible and whether these algorithms really reduce loss interference between different tasks. As such, our study contributes to the growing body of literature that aims to provide a reality check on recent algorithmic proposals in the machine learning community [11, 12, 23, 28]. We provide the following observations:

- Despite the added complexity, MTO algorithms fail to improve the interference profile beyond what is achievable by simple static weighting of the tasks (Section 4).

- The performance of multi-task models is sensitive to basic optimization parameters such as learning rate and weight-decay. Insufficient tuning of these hyper-parameters in the baselines, along with the complexity of evaluating multi-task models, can create a false perception of performance improvement (Section 4).

- In some instances, the gains reported in the MTO literature are due to flaws in the experimental design. Often times these reported gains disappear with better tuning of the baseline hyperparameters. In addition, in a handful of cases, we were unable to reproduce the reported results (Section 4.2).

- Finally, we discuss the implications for the community and the potential steps that need to be taken to standardize evaluation for multi-task models (Section 5).

## 2  Setting

We focus our discussion on the supervised learning setup, where the model parameters, $\boldsymbol{\theta} \in \mathbb{R}^p$, are trained on $K$ different tasks. We denote the loss associated with task $i$ with $\mathcal{L}_i(\boldsymbol{\theta})$.

For some problem instances, the parameter space contains a globally optimal point that achieves the best possible performance on all tasks. Figure 1 (left) provides a cartoon example of one such scenario. However, for most realistic setups, a globally optimal $\boldsymbol{\theta}$ doesn't exist. In these cases, different tasks compete with each other for model capacity. In these scenarios, the concept of Pareto optimality is used to capture the optimal trade-off in between the tasks:

**Definition** (Pareto Optimality). *$\boldsymbol{\theta} \in \mathbb{R}^p$ Pareto dominates another $\boldsymbol{\theta}'$ if $\forall 1 \leq i \leq K$, $\mathcal{L}_i(\boldsymbol{\theta}) \leq \mathcal{L}_i(\boldsymbol{\theta}')$ and there exists a task $j$ where $\mathcal{L}_j(\boldsymbol{\theta}) < \mathcal{L}_j(\boldsymbol{\theta}')$. $\boldsymbol{\theta}$ is Pareto optimal if it is not dominated by any other point. The collection of the Pareto optimal points is denoted as Pareto front.*

Figure 1 (center) provides a cartoon representation of the Pareto front for a two-task setup. The Pareto front represents the collection of parameters that achieve the best possible trade-off profile between the tasks. A practitioner can aim to land on a particular point on this trade-off curve depending on their (implicit or explicit) utility function. The location and the curvature of the Pareto curve represent the severity of the interference problem. Ideally, one would like to identify training protocols that push the trade-off curve towards the origin as much as possible (Figure 1-right).

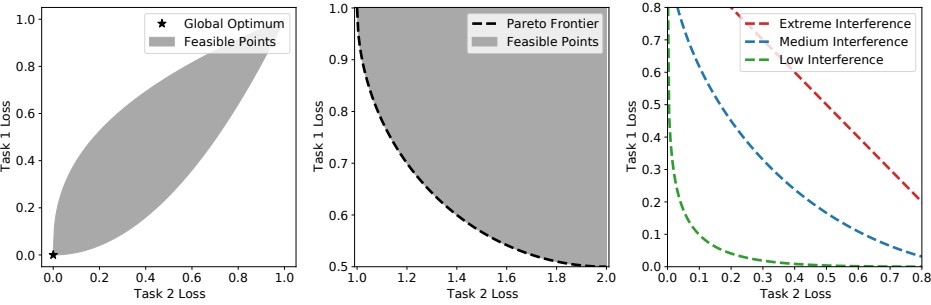

Figure 1: A cartoon representation of the achievable trade-offs in a two-task setup.

The traditional approach to optimize multi-task models is via *scalarization* [3]:

$$\hat{\boldsymbol{\theta}}(\boldsymbol{w}) = \arg\min_{\boldsymbol{\theta}} \mathcal{L}(\boldsymbol{\theta}; \boldsymbol{w}) \quad \text{where} \quad \mathcal{L}(\boldsymbol{\theta}; \boldsymbol{w}) \equiv \sum_{i=1}^{K} \boldsymbol{w}_i \mathcal{L}_i(\boldsymbol{\theta}), \quad \boldsymbol{w} > 0, \quad \sum_i \boldsymbol{w}_i = 1. \quad (1)$$

Here, $w$ is a fixed vector of task weights determined by the practitioner beforehand. The algorithmic and computational simplicity of this approach has made scalarization highly popular in practice.

Scalarization comes with certain theoretical guarantees. It can be easily shown that any solution to problem (1) is guaranteed to be Pareto optimal. In addition, when $\{\mathcal{L}_i\}_{i=1}^K$ are convex, there exists a partial converse:

**Theorem** (Informal). *Let $\theta^\#$ be a point on the Pareto front. Then there exists $w^\# \geq 0$ such that scalarization with $w^\#$ yields $\theta^\#$.* [2]

These results suggest that, at least for convex setups, sweeping the task weights should be sufficient for full exploration of the Pareto frontier. In particular, **in the convex setting it is provable that no algorithm can outperform properly chosen scalarization that has been trained to convergence.**

The results above raise a series of questions. Where are the reported improvements of MTO algorithms coming from? Is non-convexity adding additional complexity which makes scalarization insufficient for tracing out the Pareto front? Is it the case that neural networks trained via a combination of scalarization and standard first-order optimization methods are not able to reach the Pareto Frontier? Do MTO algorithms achieve a better performance trade-off curve? In following sections, we empirically examine these questions for several popular deep learning workloads.

## 3 Prior Work

There has been a flurry of research on MTO algorithms over the past few years. [13, 4] argue that finding the appropriate scalarization weights is often costly. To alleviate this, they provide algorithms that aim to automatically find a reasonable set of task weights. Sener & Koltun (2019) [26] approach multi-task learning from a multi-objective optimization view point and suggest Multiple Gradient Descent Algorithm (MGDA) for efficiently finding Pareto optimal solutions. [18, 21, 30, 31] hypothesize that negative interactions between the gradients of different tasks is a significant contributor to the interference problem. As such, these studies put forward various suggestions for projecting out conflicting gradients in order to improve the optimization dynamics. Finally, [5, 20] propose algorithms that inject randomization into the training pipeline and argue that this added randomness improves the training dynamics by allowing the optimization trajectory to escape poor local minima.

It is important to note that MTO algorithms often come with substantial computational overhead. Chen et al. (2020) report 2-5 fold increase in the training time for a 40-task benchmark [5]. Similarly, Kurin et al. (2022) observe that on some benchmarks MTO algorithms can train as much as $35$ times slower compared to scalarization [16].

Recently, there has been a number of studies that critically question the benefits of MTO algorithms. The closest such study to ours is Kurin et al. (2022) [16] that appeared on Arxiv during the preparation of this manuscript. The paper argues that MTO algorithms implicitly regularize the model and shows that with careful regularization, scalarization with equal weights can match the performance of MTOs on various popular benchmarks.

In contrast, we argue that MTO algorithms yield different solutions on the same trade-off curve (See Figure 2 for an example). In most cases, these solutions tend to be different from equal weighting scalarization solution. When performance on popular benchmarks is concerned, we argue that scalarization baselines are often under-tuned. With additional tuning of the hyper-parameters, we find that most optimizers yield comparable results.

## 4 Experiments

### 4.1 Multilingual Machine Translation

In this section, we examine the effect of MTO algorithms on multilingual neural machine translation (NMT). In particular, we focus on translation out of English as prior work has reported significant task interference in this translation direction [1].

---

[2]The precise statements and their proofs are provided in the appendix.

We start off by examining models trained jointly on English→{French, Chinese} translation tasks. The two-task setup allows us to effectively visualize the performance trade-off curves. French and Chinese are specifically chosen due to the large difference in their semantic and syntactic structures. Here, we anticipate a large degree of interference among the tasks—a setting where MTOs claim to improve upon scalarization. We repeat our experiments for English→{French, German} and English→{French, Romanian} translation tasks to ensure that our observations generalize across different task setups with different levels of data imbalance. See Table 1 for an overview of data sources. All models use (pre-LN) Transformer architecture [29] and have been trained using early stopping. See Appendix A for training details.

Table 1: Overview of data sources used in our NMT experiments.

| Language Pair | Dataset | # Train Examples | # Eval Examples |
|---|---|---|---|
| English-French | WMT15 | $40,853,298$ | $4,503$ |
| English-Chinese | WMT19 | $25,986,436$ | $3,981$ |
| English-German | WMT16 | $4,548,885$ | $2,169$ |
| English-Romanian | WMT16 | $610,320$ | $1,999$ |

We compare the performance trade-offs achieved by various popular MTO algorithms with the Pareto frontier of scalarization. Following the NMT literature's convention, we implement scalarization via proportional sampling. Here, the average number of observations in the batch corresponding to task $i$ is proportional to $\boldsymbol{w}_i$. In this setup, the expected training loss is equal to

$$\mathcal{L}(\boldsymbol{\theta}) = \mathbb{E}_{\boldsymbol{x}}[\ell(\boldsymbol{x}; \boldsymbol{\theta})] = \sum_{i=1}^{K} \mathbb{P}(\boldsymbol{x} \in \text{task } i)\mathbb{E}_{\boldsymbol{x}}[\ell(\boldsymbol{x}; \boldsymbol{\theta})|\boldsymbol{x} \in \text{task } i] = \sum_{i=1}^{K} \boldsymbol{w}_i \mathcal{L}_i(\boldsymbol{\theta}).$$

We compare scalarization with a series of popular MTO algorithms: Multiple Gradient Descent (MGDA) [26, 7, 18], GradNorm [4], Gradient Surgery (PCGrad) [31], IMTL [21], and Random Loss Weighting (RLW) [20]. For GradNorm's $\alpha$ hyper-parameter, we perform a grid search and report all non-Pareto dominated models. To give an apples-to-apples comparison, all models have been trained with the same batch-size for the same number of training steps. All models use Adam [14] as the base optimizer. For all these optimizer categories, we tune the learning rate on a grid from $5 \times 10^{-2}$ to $5$ and report all non-Pareto dominated models. Details of the training and hyper-parameters are presented in Appendix A.

The overview of our experimental findings are presented in Figures 2, 4, and 5. The blue dashed line corresponds to the Pareto front achieved via proportional sampling with English→French sampling rate ranging from 10% to 90%. Several observations are in order:

**No Improvements from MTO Algorithms**   Despite the promise to alleviate interference among the tasks, all of the MTO algorithms in our study simply yield performance trade-off points on the scalarization Pareto front. As such, their performance can be fully replicated by simply optimizing a weighted average of the losses. To understand this phenomenon better, in Figure 3, we plot the evolution of the task weights for PCGrad, MGDA, GradNorm, and IMTL during training. We observe that for the majority of the training runs, the dynamically assigned task weights do not move significantly. As such, in effect, these MTO algorithms behave similar to static weighting.

**Other Language Pairs**   En→Fr and En→Zh are both high-resource tasks with $O(10^7)$ training examples. In these experiments, we observe minimal overfitting and excellent agreement between train and test behaviors. One might argue that MTO algorithms possess transfer learning and regularization capabilities En→{Zh, Fr} experiments downplay [3]. To address this, we repeat our experiments in two new task setups where En→Zh task is substituted with En→De (mid-resource) and En→Ro (low-resource).

Figures 4 & 5 present the results of these experiments. En→{De, Fr} experiments closely resemble En→{Zh, Fr} ones: MTO algorithms simply achieve different trade-off points on the scalarization Pareto front. The results for En→{Ro, Fr} are more interesting. We still observe a clear Pareto

---

[3]See [16] for an overview of proposals on how MTO algorithms can perform implicit regularization.

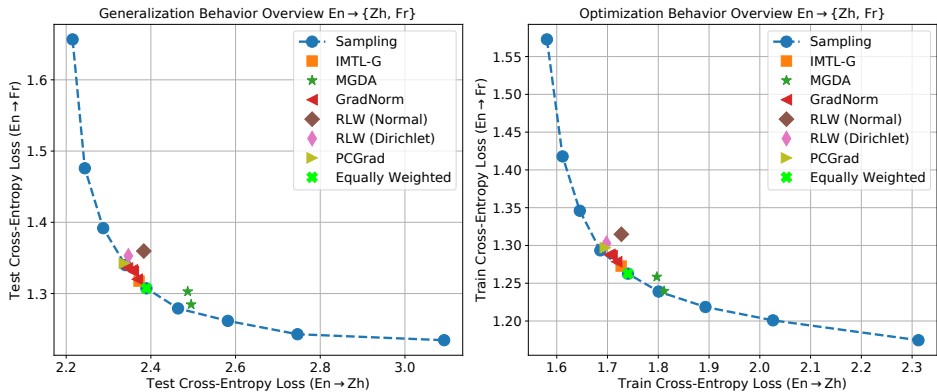

Figure 2: Performance trade-off behavior for En→{Fr, Zh} models. Each point corresponds to the final performance of a model. We observe no improvements in terms of final performance or training behavior from MTO algorithms. See Appendix B for more comparisons.

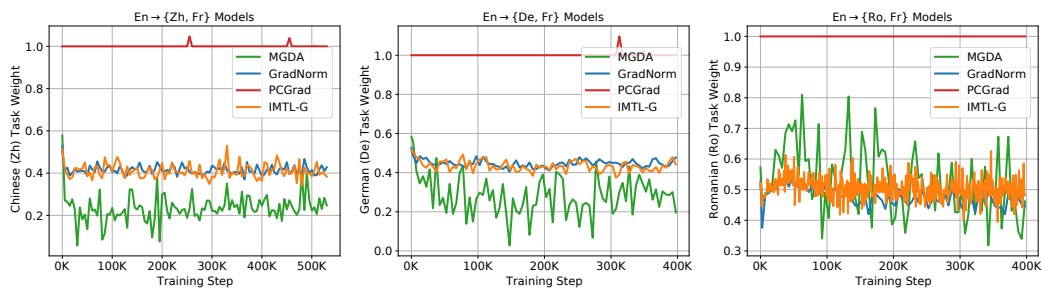

Figure 3: The evolution of task weights during training. All models are trained with the same (near optimal) base learning rate of $0.5$. For the majority of the runs, the task weights barely move.

front for the training performance; different MTO algorithms achieve different points on this curve. For the generalization behavior however, the Pareto frontier ceases to exist. Instead, we observe models that are *globally optimal*. Interestingly, these globally optimal solutions are found only by scalarization (with sampling rates close to $(0.3, 0.7)$). In contrast, MTO algorithms find solutions with near equal task weights which yield significantly worse generalization performances. As the generalization performance in this setup is primarily driven by the amount of regularization applied to the low-resource task during training, our results cast doubt on the ability of MTO algorithms to effectively regularize the model.

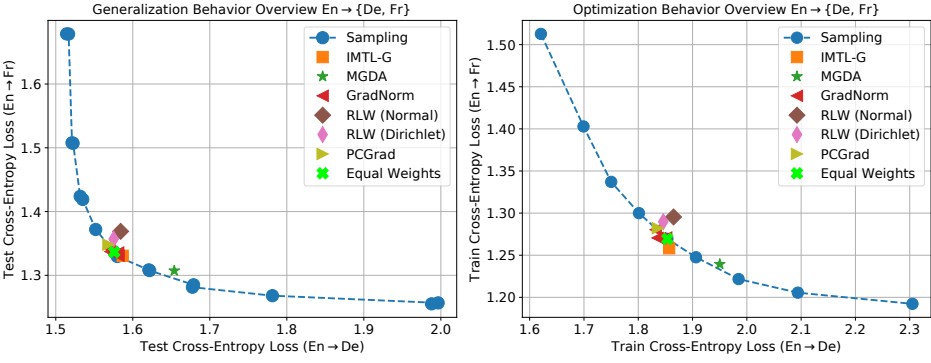

Figure 4: Performance trade-off behavior for En→{De, Fr} models. The results closely resemble our observations on En→{Zh, Fr} experiments.

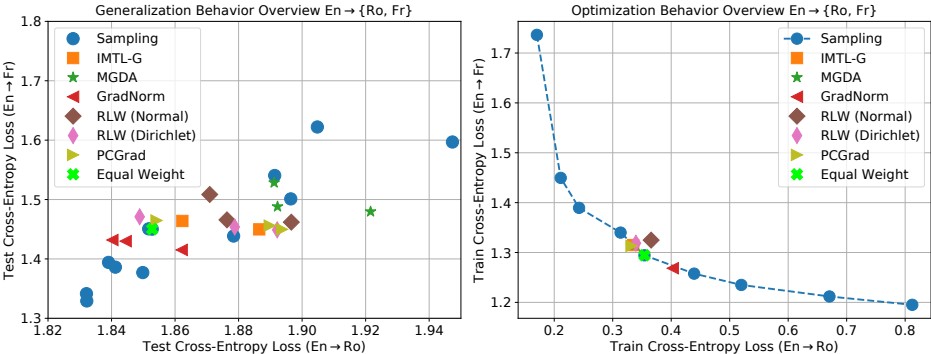

Figure 5: Performance trade-off behavior for En→{Ro, Fr} models. *Left:* We do not observe a Pareto frontier for the test performance. Instead, scalarization with weights $(0.3, 0.7)$ achieves a globally optimal trade-off and outperforms MTO algorithms. *Right:* For the training loss, a clear Pareto frontier appears and MTO algorithms simply selects a point on the performance profile traced out by the scalarization sweep.

**Evaluation Challenges**    Our experiments suggest that the model performances are highly sensitive to the choice of hyper-parameters. Even subtle choices regarding the hyper-parameter grid can drastically change the results. For example, it is common practice to tune the learning rate on a sparse grid, say sweeping $\eta \in [10^{-3}, 10^{-2}, 10^{-1}]$. How much do our reported metrics suffer from such a sparse sweep, and how much performance can be gained on average from further tuning of just this one hyperparameter? To answer this question, we simulate running multiple instances of sparse grid searches of the form $\{k \times 10^{-3}, k \times 10^{-2}, k \times 10^{-1}\}$ for $1 \leq k \leq 9$; each choice of $k$ produces a tuning study sweeping over 3 learning rates. We then measure the variance in the optimal performance of each 3 trial study, as $k$ is varied from 1 to 9. The results are shown in Figure 6 (left). For comparison, we plot the variance in performance resulting from running the best $\eta$ multiple times with different seeds. Notably, the effective standard deviation resulting from sparse learning rate tuning is 6 to 7 times the standard deviation observed from varying the random seed for a fixed hyperparameter point. The upshot is, **estimating trial variance by rerunning multiple seeds is insufficient for concluding that performance gains from a new algorithm are significant when the hyperparameters are sampled on a sparse grid**.

The established convention in the literature for ranking MTO performances is to compare some kind of average of the per-task performances. The specific average used is fixed somewhat arbitrarily for the purposes of benchmarking. However, in practice, the utility function for ranking algorithms may vary dramatically depending on the goals of the practitioner. Thus, useful MTOs need to be robust to changes in the utility function. Either they need to improve upon the performance profile curve traced out by a sampling sweep, or reliably find better points on the profile curve with minimal tuning as the utility function is varied. Unfortunately, the current practice to consider just one (arbitrary) weighting scheme will bias the evaluation towards algorithms that perform well on that specific scheme but are not robust to changes in the utility function. For example, Figure 6 (right) ranks 3 MTOs as the evaluation per-task weighting is varied. As the En→Zh weight is varied, the ranking shifts from MGDA being the best MTO to PCGrad being the best. This is a natural consequence of Figure 2 which shows that different MTOs find different points on the same Pareto front traced out by a sampling sweep. Notably, no algorithm outperforms sampling with a well-chosen sampling ratio.

**Alternative Approaches**    As discussed in Section 3, MTO algorithms often drastically increase the algorithmic and computational complexity of the training process. In our experiments, we observed that the requirement to compute per-task gradients (which is necessary for many MTO algorithms) led to a significant reduction in the number of training steps per second (from $\approx 12$ to $\approx 5$). Given these observations, it is natural to wonder if there are more effective ways to spend this extra compute budget. Figure 7 examines how scaling the model size changes the performance trade-off behavior. We examine increasing the model depth by a factor of $\{1, 2, 3, 4\}$. Our largest model achieves on average $5.4$ training steps per second, which is comparable with the models trained using per-task gradients. Our results suggest that, unlike the observed behavior with MTO algorithms, allocating

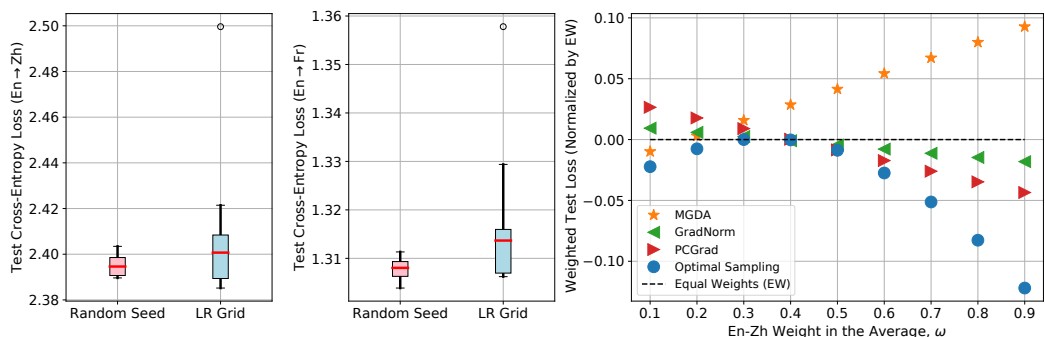

Figure 6: *Left:* Sparse sampling of learning rates has a significantly larger effect on reported performance than varying the random seed of a particular hyperparameter point. *Right:* Rankings between algorithms can depend on how tasks are weighted at evaluation time. For example, if the eval performance is ranked according to .9 En-Fr + .1 En-Zh, then MGDA is preferred to GradNorm and PCGrad. At equal weighting, GradNorm and PCGrad outperform MGDA. For any weighting, all MTOs underperform an optimally chosen sampling scheme.

more compute to scale the model yields consistent improvements across the board. Larger models achieve Pareto fronts strictly to the lower-left of the base model, which corresponds to a performance improvement for all utility functions.

Table 2: Overview of models used in the scaling experiment.

| Model | Optimizer | # Parameters | # Steps/Sec |
|---|---|---|---|
| 3L-3L (base) | Scalarization | 120M | 11.52 |
| 3L-3L | MGDA | 120M | 4.81 |
| 6L-6L | Scalarization | 142M | 8.28 |
| 9L-9L | Scalarization | 165M | 6.68 |
| 12L-12L | Scalarization | 187M | 5.48 |

## 4.2 Benchmarks from the Literature

The observations of Section 4.1, run contrary to many recent influential studies proposing MTOs for multi-task models [4, 5, 9, 26, 31]. These papers often compare the performance of their proposed algorithm with traditional training strategies and report significant gains. In this section, we attempt to reproduce these results on a number of supervised-learning benchmarks. We present comparisons for CityScapes [6] and CelebA [22] datasets in the main text.[4]

For these experiments, we closely follow the experimental setup and the publicly available code from [26]. We modified the code sparingly to address bugs, update deprecated libraries, and speed up the data loader. We perform an extensive grid search for learning rate, weight decay, and dropout. All models use early stopping. Our implementation details are presented in the appendix.

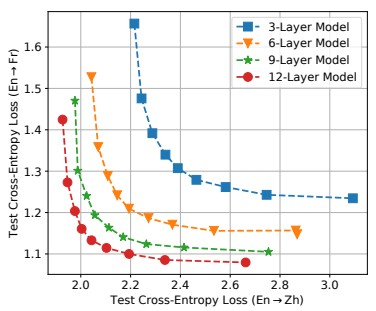

Figure 7: The effect of model size on the Pareto frontier for En→{Zh, Fr} models.

---

[4]See Appendix C for more comparisons and details.

### 4.3 CityScapes

CityScapes [6] is a dataset for understanding urban street scenes. It is constructed via stereo video sequences from different cities and contains 2975 training and 500 validation images. In the multi-task optimization literature, this dataset is popularly cast as a two-task problem with one task being 7-class semantic segmentation and the other being depth estimation. In our experiments, we choose 595 random samples from the training data to serve as our validation set. This validation set is used for tuning hyper-parameters such as learning rate and weight decay (See appendix for details). We use the original validation set as our test set.

Figure 8 provides an overview of our experimental results. Similar to Section 4.1, we observe that scalarization solutions form the generalization performance Pareto front. This frontier is observable both for test loss (left) and task specific generalization metrics (right). In both cases, MTO solutions significantly under-perform scalarization.

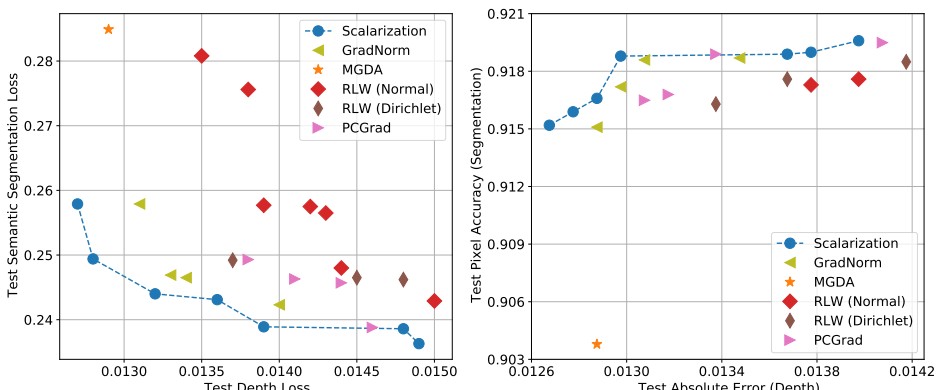

Figure 8: The generalization performance of different optimizers for CityScapes benchmark. *Left:* Test segmentation loss vs test depth loss. Points on the lower left side represent better solutions. *Right:* Test pixel accuracy vs test absolute depth estimation error. Here, points on the upper left side are better solution. In both cases, scalarization solutions form the Pareto frontier.

For CityScapes models, the segmentation task loss is an order of magnitude larger than the depth estimation task loss. This severe loss imbalance causes interesting behaviors to emerge that are worth noting. Figure 9 examines the train / test behavior of the different scalarization solutions. Contrary to recent results reported in the literature [16], we observe that appropriately balancing the different losses is crucial in achieving a desirable generalization behavior: in Figure 9, the majority of the generalization Pareto frontier is populated by models with segmentation task weight less than 0.2.

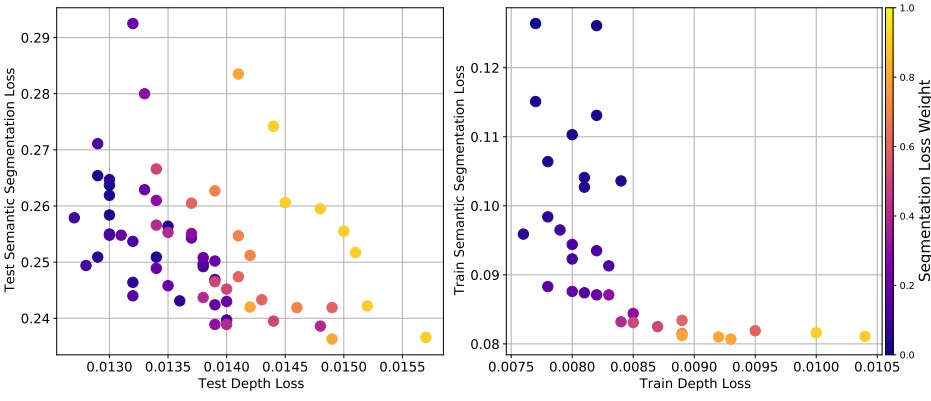

Figure 9: Scalarization test (*left*) and train (*right*) Pareto frontiers for CityScapes dataset. We plot all non-dominated experimental runs per scalarization weight mixture.

### 4.3.1 CelebA

CelebA dataset [22] is a collection of 200K face images annotated with 40 attributes. This dataset is a popular benchmark for MTO research where each attribute is treated as a separate binary classification task. In Section 4.1, we identified a number of evaluation challenges for MTO algorithms, namely the significance of exact hyper-parameter tuning and the difficulty of comparing models via average performances. These evaluation challenges become highly visible for CelebA.

Figure 10 presents the overview of our results. We report average performance across the tasks. Our results suggest that scalarization performance is comparable with the performance of popular MTO algorithms. This is in line with recent findings in the literature [16]. More importantly, Figure 10 shows the importance of careful hyper-parameter tuning: Even in the presence of early-stopping, there is significant variation in the final performance of the models that is drastically larger than the effect of the MTO algorithm choice.

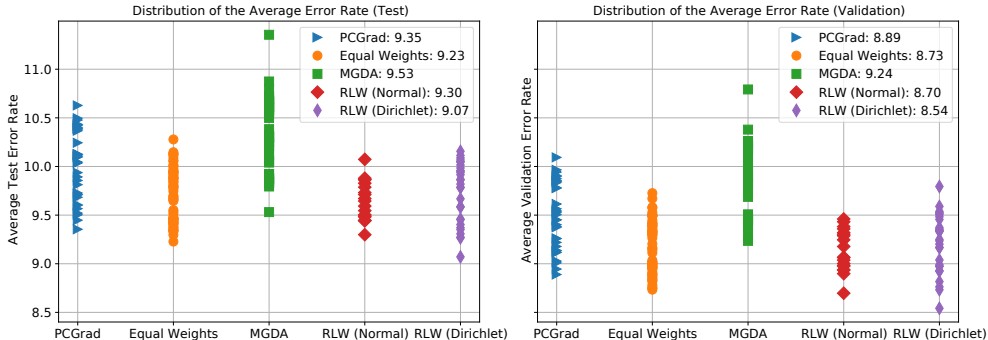

Figure 10: For CelebA, the effect of tuning hyper-parameters is much more significant than the effect of the MTO algorithm choice. Each point here corresponds to the performance of an early-stopped model. We vary the learning rate from $10^{-4}$ to $5 \times 10^{-1}$ and weight decay from 0 to $5 \times 10^{-3}$.

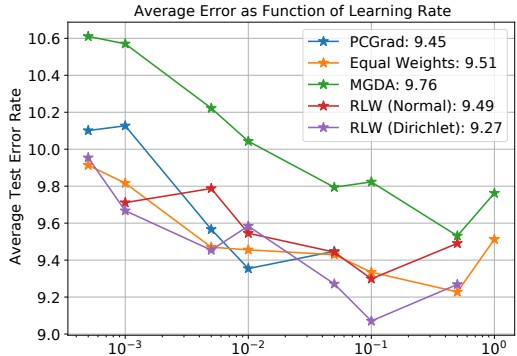

Figure 11: Average test error as a function of learning rate for each MTO on CelebA. The performance ranking of the different MTOs is highly dependent on the learning rate. Due to this high variability, studies with sparse learning rate grids can yield misleading conclusions.

This acute hyperparameter sensitivity can lead to a scenario where insufficient tuning of baseline hyper-parameters gives illusions of significant performance gains. We suspect such evaluation challenges play a prominent role in the significant disagreements we observe in the literature on the effect and ranking of MTO algorithms. Table 3 presents an overview of the results presented in the literature. As the table suggests different papers report wildly different performance for the same algorithm. A large fraction of the reported statistics resemble the quantities we observe on the *validation* dataset. This is concerning as validation performance on CelebA tends to be noisy. If only the validation performance is reported, a combination of early stopping and high evaluation frequency can artificially boost the scores. This artificial boost in scores is clearly visible between the left and the right sides of Figure 10.

Table 3: An overview of reported results in the literature for CelebA benchmark. There is significant disagreement between reported statistics from different studies. We suspect improper tuning of baseline hyper-parameters is a likely culprit (compare reported statistics with Figure 10).

| Study | [5] | [26] | [31] | [21] | [16] | Ours (Test) | Ours (Validation) |
|---|---|---|---|---|---|---|---|
| Scalarization | 8.71 | 9.62 | – | 9.99 | 9.1 | 9.23 | 8.73 |
| MGDA | 10.82 | 8.25 | 8.95 | 9.96 | 9.78 | 9.53 | 9.24 |
| GradNorm | 8.68 | 8.44 | – | 10.08 | – | – | – |
| PCGrad | 8.72 | – | 8.69 | 10.01 | 9.07 | 9.35 | 8.89 |
| GradDrop | 8.52 | – | – | – | 9.02 | – | – |

## 5   Conclusions

In this paper, we presented a large-scale empirical study examining the effects of multi-task optimization methods. It is often assumed that these algorithms enhance the optimization dynamics of multi-task models and yield desirable solutions that cannot be achieved via scalarization. Our results suggest the contrary. Across a variety of language and vision tasks, we showed that scalarization, with appropriate weights, can match both the optimization and the generalization behaviors of MTO algorithms. As such, in effect, scalarization solutions form a superset for MTO solutions. Our experimental results suggest effective exploration of the scalarization solution set might be a more reliable and effective strategy for boosting the model performance (See Fig 5).

Our observations suggest the final performance of multi-task models is highly sensitive to the choice of training hyper-parameters. Often times, the effect size associated with subtle design decisions in the choice of the hyper-parameter grid is orders of magnitude larger than the MTO effect size (See Fig 10). As such, researchers can unknowingly create the illusion of significant performance gains by simply under-tuning the competing baselines. The fact that different studies are reporting drastically different numbers for the same dataset-algorithm pair (Table 3) suggests this phenomenon is prevalent in this literature.

**Limitations and Future Research**   Our results suggest that by exploring the scalarization solution space, one can attain performance on par with (or better than) many MTO algorithms. However, the grid search approach we used for computing the scalarization Pareto frontier is computationally prohibitive. Examining strategies for efficiently searching this solution space (such as [10, 15]) is a fruitful future research direction.

In the paper, we pointed out worrying concerns regarding faulty evaluation and under-tuned baselines. A natural solution to alleviate these problems is to adopt the Common Task Framework (CTF) [8, 19] to reliably identify and measure algorithmic improvements in multi-task optimization. With the creation of a commonly used competitive benchmark with a proper validation / test split, baselines will naturally become stronger as subsequent papers progressively improve performance—this makes substantial gains more convincing than current practice where baselines rerun by the authors themselves. We postpone the development of such pipeline to future work.

Finally, to keep the discussion tractable, we focused our analysis to supervised learning benchmarks. Whether the same behavior holds for reinforcement learning and self-supervised learning setups is still an open question.

## Acknowledgments and Disclosure of Funding

We thank George E. Dahl, Wolfgang Macherey, and Macduff Hughes for their constructive comments on the initial version of this manuscript. Additionally, we thank Sourabh Medapati and Zachary Nado for their help in debugging our code base. Moreover, we are grateful to Soham Ghosh and Mojtaba Seyedhosseini for valuable discussions regarding the role of MTOs in large-scale models.

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
