# A    NMT Training Setup

In this appendix, we provide full details of our experimental setup for Section 4.1. All models use pre-LN encoder-decoder transformer architecture. The base model, used for the majority of the experiments of this section, has 3 encoder layers and 3 decoder layers. Note that we intentionally chose a small model to exacerbate interference among the tasks and make our experimental setup more favorable to MTO algorithms. Following the NMT literature convention, our models are trained with $0.1$ label smoothing and $0.1$ dropout [27] for feed-forward and attention layers. We use a sentence piece vocabulary of size $64K$ for our models. Table 4 provides the architecture details.

Table 4: Overview of network and optimizer hyper-parameters.

| Hyper-parameter | |
|---|---|
| Feed-forward dim | 2048 |
| Model dim | 512 |
| Attention heads | 8 |
| Attention QKV dim | 512 |
| Label smoothing | 0.1 |
| Dropout | 0.1 |
| Batch-size | 1024 |
| Warm-up steps | 40K |

Models are trained using Adam optimizer [14] with a fixed batch-size of 1024. En→{Zh, Fr} models are trained for 530038 steps while the rest of models (due to smaller training data size) are trained for 397529 steps. For all the runs, we use $40K$ steps of linear warm-up and then use a learning rate schedule of the form

$$\frac{\eta}{\sqrt{t}}, \qquad \eta : \text{base learning rate}, \qquad t : \text{training step}.$$

For each model run, we sweep for $\eta$ in the grid $\{0.05, 0.1, 0.5, 1.0, 2.5, 5.0\}$. Often times, $\eta = 0.5$ yields the optimal performance and $\eta = 5.0$ diverges. For sampling experiments, we sweep the rate for En→Fr in the grid $\{i/10\}_{i=1}^{9}$. This determines the rate for the other language pair automatically. As such, to derive each scalarization front, we train a total of $54$ models.

Some of the MTO algorithms under our investigation have algorithm-specific hyper-parameters. In particular, RLW [20] requires specifying the task weight distribution and GradNorm [4] requires specifying a parameter $\alpha$. For RLW, we examined Gaussian and Dirichlet distributions and presented the results separately in our plots. For GradNorm, we sweep for $\alpha$ in the grid $\{0.25, 0.5, 0.75, 1.0, 1.25, 1.5\}$ and present all non-Pareto dominated models.

When examining the generalization performance (left hand side of Figures 2, 4, and 5) we use early stopping: we evaluate the model every 5000 steps and use the step that yields the smallest average validation loss for the two tasks. For En→{Zh, Fr} and En→{De, Fr} models, it is often the case that the final step is the optimal step. As such early stopping doesn't significantly change the picture. For En→{Ro, Fr}, performance statistics change noticeably with early stopping but the overall qualitative picture remains the same. For the training performance (right hand side of Figures 2, 4, and 5) we report the final step training statistics.

# B  Additional Results

In this appendix section, we provide additional performance comparisons for NMT models trained in Section 4.1.

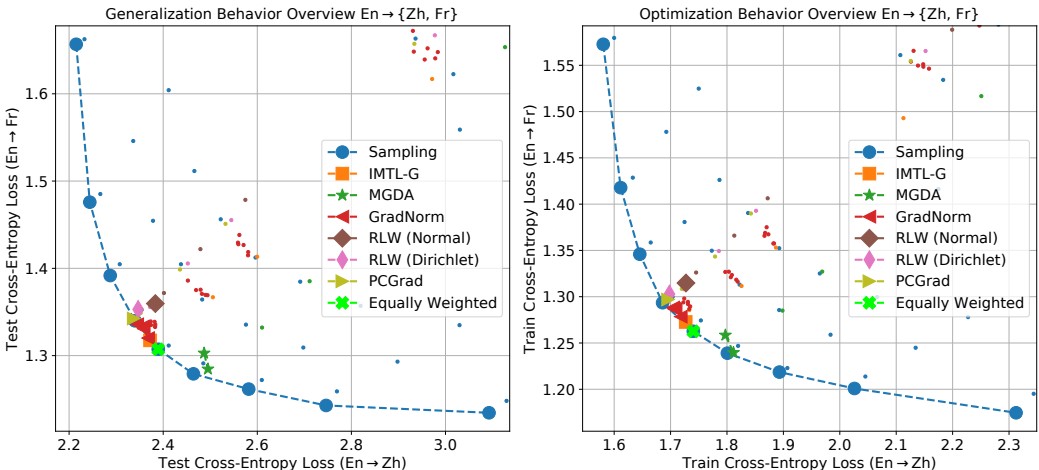

Figure 12: The full generalization / optimization performance overview for En→{Zh, Fr} models. Small dots correspond to Pareto dominated models excluded from Figure 2 to avoid clutter. Pareto dominated trade-off curves correspond to models trained with suboptimal base learning rate.

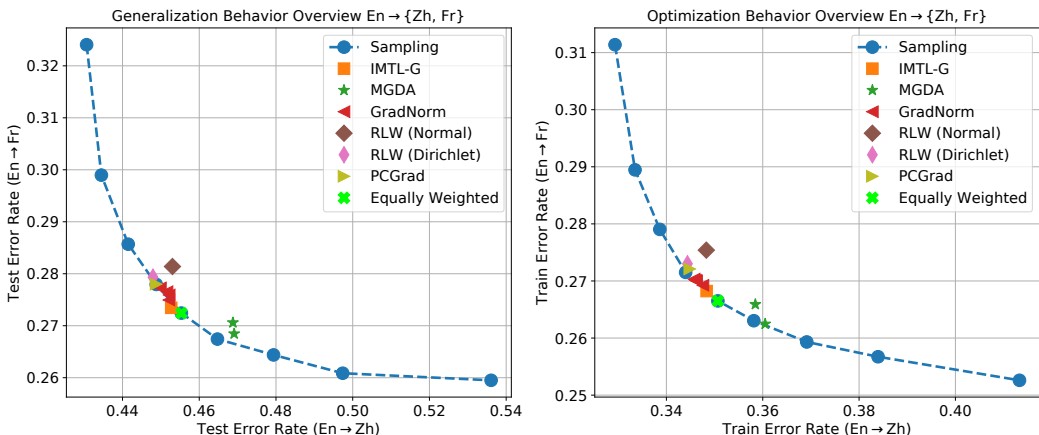

Figure 13: Observations of Section 4.1 generalize across the choice of performance metrics. *Left:* Next token prediction error rate evaluated on the validation data. *Right:* Next token prediction error rate evaluated on the training data.

In order to avoid the artifacts and complexities decoding, in the main text, we used cross-entropy loss as the main evaluation metric for models in Section 4.1. To complete the picture, Figure 14 examines the quality of generated translations (as measured by (Sacre-)BLEU score [24, 25]). All translations are generated via Beam-Search with beam size of 4. Note that, for the sake of computational tractability, we do not optimize the decoding algorithm hyper-parameters for each model. As such the performance trade-off frontier is more noisy.

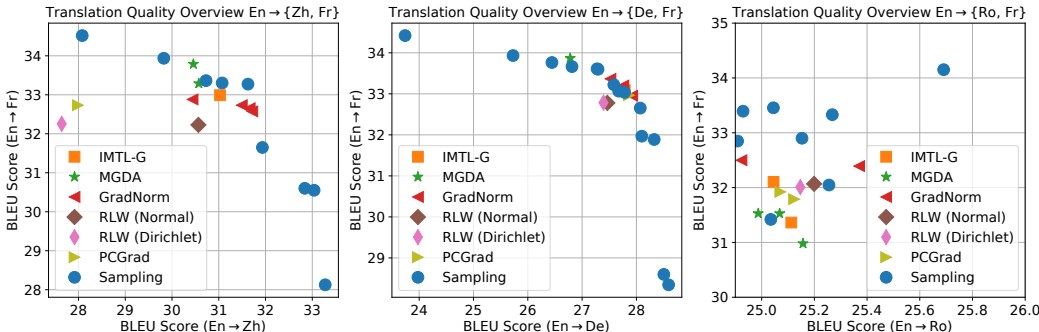

Figure 14: Translation quality of our models as measured by BLEU score. For En→{Ro, Fr} models (right) scalarization clearly outperforms the rest of the optimizers.

# C  Vision Benchmarks

We analyze results on three main vision benchmarks used in multi-task optimization, Multi-MNIST [26], CelebA [22] and CityScapes [6]. Multi-MNIST is a two task dataset, which uses the handwritten digits of MNIST but overlays a right digit and a left digit over each other. CelebA is a dataset of celebrity faces and is cast as a 40-task classification problem; each task predicts a different attribute of the face. Finally, CityScapes is a dataset for understanding urban street scenes. In our setting, it is a two task problem with one task being 7-class semantic segmentation and the other being depth estimation.

We would like to thank Lin et al. (2021) [20] and Sener et al. (2018) [26] for publicly releasing their code. Our CelebA and Multi-MNIST experiments heavily utilize code from Sener et al. and our CityScapes experiments heavily utilize code from Lin et al. For CelebA and Multi-MNIST, our primary changes to the code base include integrating more optimization algorithms, speeding up the dataloaders via the Tensorflow datasets library and creating a validation set for Multi-MNIST by partitioning the training set. Our validation set for Multi-MNIST is 12000 images, while our training set is 48000 images. We use the original MNIST testing set as our test set, but transformed to a multi-task setting. For CityScapes, we primarily changed the dataloader to have it pre-load images into memory, added statistic tracking for the validation set, and integrated other optimizers.

## C.1  Hyper-Parameter and Experiment Details

**Multi-MNIST**  For all optimizers, we searched through all combinations of learning rate $\eta \in [0.001, 0.005, 0.01, 0.05, 0.1, 0.5, 1.0, 5.0]$, and dropout rate $\gamma \in [0.1, 0.2, 0.3, 0.4, 0.5]$. We use a LeNet architecture detailed in Sener et al. (2018) [26] with two fully-connected layers devoted for each task. For GradNorm specifically, we also search through $\alpha \in [0.5, 1.0, 1.5, 2.0]$. Our learning rate follows a step-wise scheduler with a multiplicative factor of $0.85$ every 30 epochs. To create our dataset, we follow steps outlined in Sener et al. (2018), overlaying two random digits on top of each other, one positioned at the top left, and the other at the bottom left. We then resize the image to $28 \times 28$. We use batch size of 256 and SGD with momentum of $0.8$.

**CelebA**  Similarly our hyper-parameter search for CelebA included all combinations of learning rate $\eta \in [0.0001, 0.0005, 0.001, 0.005, 0.01, 0.05, 0.1, 0.5, 1.0]$ and weight decay $\lambda \in [0, 10^{-5}, 5 \times 10^{-5}, 10^{-4}, 5 \times 10^{-4}, 10^{-3}, 5 \times 10^{-3}]$. For GradNorm, we search through $\alpha \in [0.5, 1.0, 1.5, 2.0]$. Our learning rate schedule was the same as the one for Multi-MNIST and we use a batch size of 256. For CelebA we also use SGD with momentum of $0.8$. The model follows the one detailed in Sener et al. (2018).

**Cityscapes**  Here our hyper-parameter search implements something slightly different. We search through all combinations of learning rates $\eta \in [10^{-5}, 10^{-4.5}, 10^{-4}, 10^{-3.5}, 10^{-3}, 10^{-2.5}, 10^{-2}]$ and weight decay $\lambda \in [0, 10^{-6}, 10^{-5.5}, 10^{-5}, 10^{-4.5}, 10^{-4}, 10^{-3.5}, 10^{-3}, 10^{-2.5}, 10^{-2}]$. For GradNorm, we search through $\alpha \in [0.5, 1.0, 1.5, 2.0]$. We use a batch size of 64 for all optimizers. We split the training data set of 2975 images into a validation set of 595 with the rest being our actual training set and we use the original validation set of 500 images as our test set. All images are resized to $128 \times 256$ and we use Adam [14] as our base optimizer. For model we use the architecture utilizing ResNet-50 as a shared encoder detailed in Lin et al. (2021) [20].

## C.2  Additional Comparisons

We present additional metrics from Section 4.2 for Cityscapes dataset. The results are presented in Figure 15. We compute mIOU for segmentation, and for depth estimation we compute absolute error. All models are trained with early stopping on validation data. The experimental results align closely with our findings in Sections 4.1 and 4.2.

In figure 16 we also present our results on the Multi-Mnist data set, whose results also align with our previous findings. We see in figure 16 the performance of MTO algorithms on this benchmark again do not out perform scalarization.

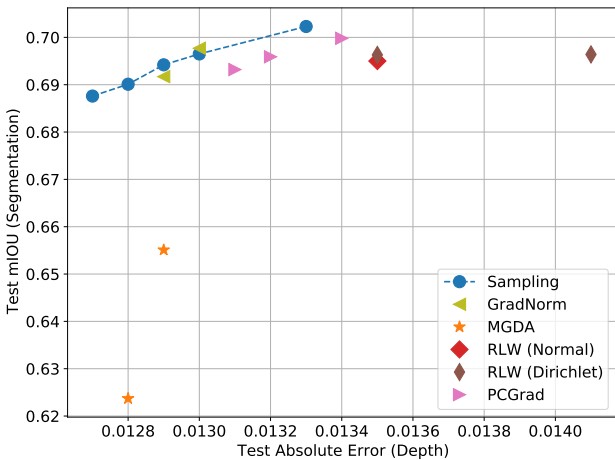

Figure 15: Additional metrics for the generalization performance of different optimizers on the Cityscapes benchmark. We have test segmentation mIOU (y-axis) and test depth absolute error (x-axis).

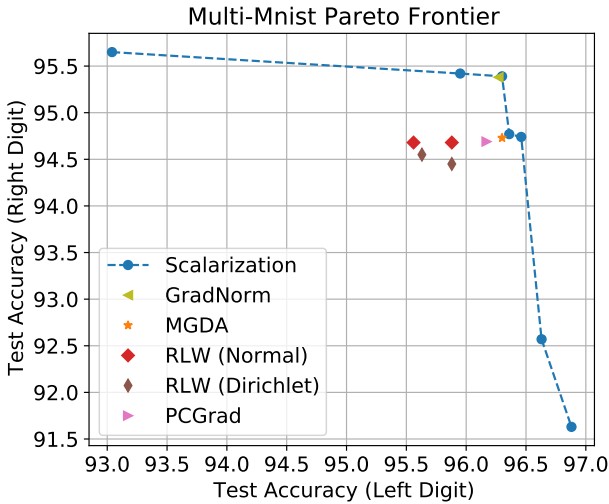

Figure 16: Test accuracy behavior on Multi-MNIST dataset aligns with our observation of Section 4.1.

### C.3 CityScape Hyper Parameter Analysis

We see more evidence that hyper parameter tuning has an immense effect on performance. Figure 17 visualizes this effect with respect to both learning rate and weight decay for each CityScape task. Secondly we note that even though hyper parameter tuning creates quite a bit of variability in performance, hyper parameters for optimally performing points are bounded within a relatively tight region. Figure 18 shows per scalarization mixture, the non pareto dominated points and their learning rate and weight decay parameters. We see that for these pareto optimal points, hyper parameters are bounded between $10^{-4}$ and $10^{-3}$ even though our search space covers the full x axis and y axis.

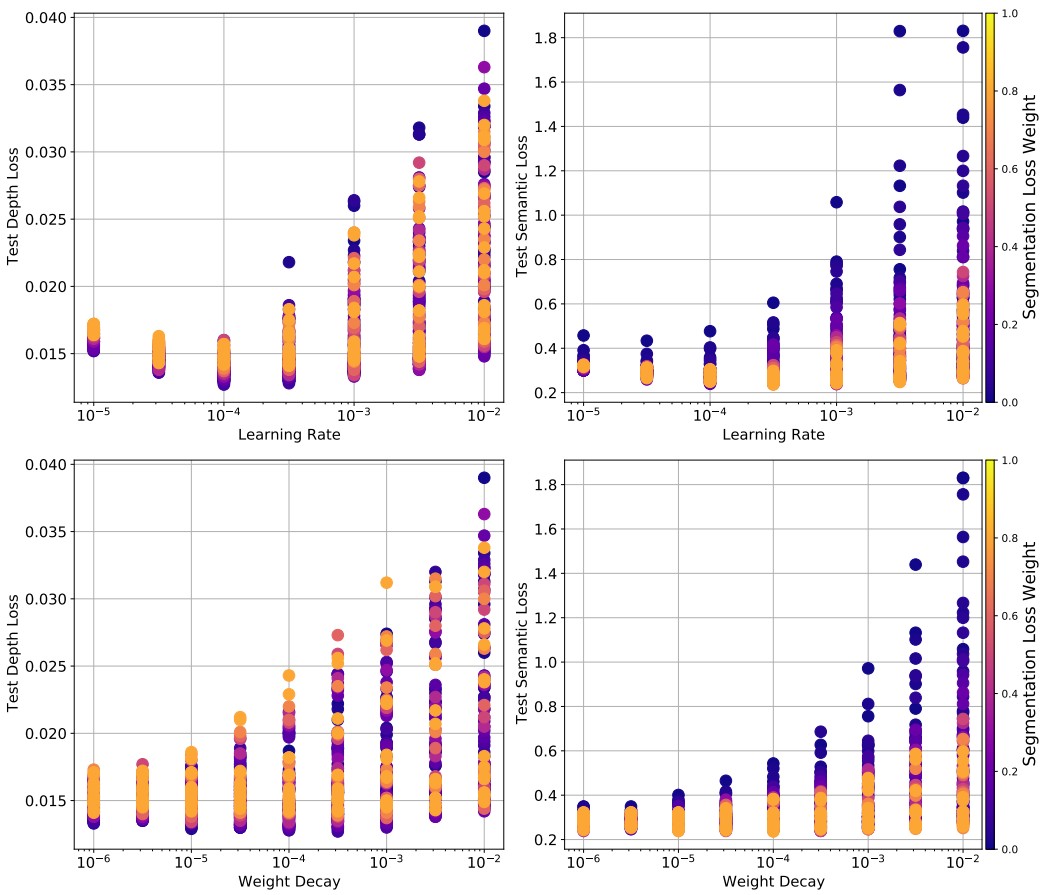

Figure 17: We visualize how sensitive the CityScape's tasks are to both learning rate and weight decay. *Top Left:* learning rate vs test depth loss. *Top Right:* learning rate vs test semantic loss. *Bottom Left:* weight decay vs test depth loss. *Top Right:* weight decay vs test semantic loss.

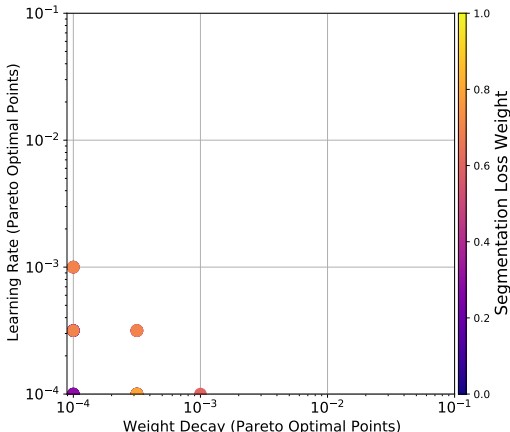

Figure 18: All points here are pareto optimal for a given scalarization mixture. Our total hyper parameter search space spans 0 to $10^{-2}$ for weight decay and $10^{-5}$ to $10^{-2}$ for learning rate.

# D  Theorem Statements and Proofs

In Section 2, we briefly discussed theoretical guarantees for Scalarization. In this appendix section, we make these statements explicit. The theorem statements and their proofs closely mirror the discussion in Section 4.7 of [3].

**Theorem 1.** *Let $\hat{\boldsymbol{\theta}}(\boldsymbol{w}) \in \arg\min_{\boldsymbol{\theta}} \mathcal{L}(\boldsymbol{\theta}; \boldsymbol{w})$ for $\boldsymbol{w} > 0$. Then $\hat{\boldsymbol{\theta}}(\boldsymbol{w})$ is Pareto optimal.*

*Proof.* Let's assume the contrary. In this case, by the definition of Pareto optimality, $\exists \boldsymbol{\theta}'$ s.t. $\forall 1 \leq i \leq K$, $\mathcal{L}_i(\boldsymbol{\theta}') \leq \mathcal{L}_i(\hat{\boldsymbol{\theta}}(\boldsymbol{w}))$ and for at least one task $j$, $\mathcal{L}_j(\boldsymbol{\theta}') < \mathcal{L}_j(\hat{\boldsymbol{\theta}}(\boldsymbol{w}))$. As such, given that $\boldsymbol{w} > 0$, we have

$$\mathcal{L}(\boldsymbol{\theta}'; \boldsymbol{w}) = \sum_{i=1}^{K} \boldsymbol{w}_i \mathcal{L}_i(\boldsymbol{\theta}') < \sum_{i=1}^{K} \boldsymbol{w}_i \mathcal{L}_i(\hat{\boldsymbol{\theta}}(\boldsymbol{w})) = \mathcal{L}(\hat{\boldsymbol{\theta}}(\boldsymbol{w}); \boldsymbol{w})$$

which contradicts our assumption that $\hat{\boldsymbol{\theta}}(\boldsymbol{w})$ is a minimizer of the problem. $\qquad\square$

**Theorem 2.** *Let $\{\mathcal{L}_i\}_{i=1}^{K}$ be convex. Also let $\boldsymbol{\theta}^{\#}$ be an arbitrary point on the Pareto frontier. Then $\exists \boldsymbol{w} \geq 0$, $\boldsymbol{w} \neq 0$ such that $\boldsymbol{\theta}^{\#} \in \arg\min_{\boldsymbol{\theta}} \mathcal{L}(\boldsymbol{\theta}; \boldsymbol{w})$.*

*Proof.* See Section 4.7.4 of [3]. $\qquad\square$

# E  Compute Resources

For the NMT experiments, we trained a total of $589$ models. Each experiment was trained on Google Cloud Platform v3 TPUs for a period of 12-28 hours. For the vision benchmarks we trained a total of $1960$ models for CityScapes, $1008$ models for CelebA, and $720$ models for Multi-Mnist. Each being trained on an Nvidia A100 GPU.