# OpenReview forum: "Do Current Multi-Task Optimization Methods in Deep Learning Even Help?"
_NeurIPS.cc/2022/Conference — NeurIPS 2022 Accept_

### Official Review · Reviewer_FSsD · 2022-07-09

**Rating:** 7
**Confidence:** 3
**Soundness:** 3 good
**Presentation:** 3 good
**Contribution:** 3 good

**Summary:**

Here, the authors present an extensive empirical evaluation of multi-task optimization (MTO) methods across Multilingual Machine Translation, Multi-MNIST and CelebA datasets. The results showed that the MTO methods tested did not offer performance improvements over scalarization as a method of optimizing MTL models. The authors also provided a comprehensive evaluation of the impact of hyperparameters. They pointed out that this and the different evaluation strategies employed by previous papers might lead to optimistic results in favor of MTO methods.

**Questions:**

Could the authors make a bit clear the limitations of this work in section five?


**Limitations:**

As per the checklist, the authors did mention limitations in Section 5. I believe the limitation is stated in the second paragraph of this section. Would it be possible for the author to clarify this? There is an abrupt transition from one paragraph to another here, and I read the paragraph twice to note the limitation.

**Strengths And Weaknesses:**

1) This work provides a comprehensive benchmarking of MTO methods using language and vision datasets. In light of the recent trend of optimization-based approaches to multi-task learning, this evaluation is of importance in the field of multi-task learning.

2) The authors presented a discussion and analysis of why previous MTO methods present optimistic results, likely due to poor fine-tuning and evaluation procedures.

3)  MTO methods might present a substantial computational overhead. The authors provided a careful analysis suggesting that time/investment in simpler and well-tuned baselines might be beneficial in the end.

---

> ### Author Response · Authors · 2022-08-02
> **Response to Reviewer FSsD**
>
> We thank the reviewer for their supportive feedback. We acknowledge the reviewer’s concern about the limitations discussion being too dispersed in the text. We have reorganized Section 5 to address these issues.

---

### Official Review · Reviewer_i2dN · 2022-07-09

**Rating:** 7
**Confidence:** 4
**Soundness:** 4 excellent
**Presentation:** 3 good
**Contribution:** 3 good

**Summary:**

This paper performs large-scale experiments to show the recent multi-task optimization methods do not improve the performance over the simple static scalarization of different tasks. The paper also shows that the performance of multi-task learning is sensitive to the chosen hyperparameters. And suboptimal hyperparameters sometimes make some MTO method stand out from others while this trend does not hold after using better hyperparameters They also suggest the potential solution as using the common task framework to make a fair comparison for multi-task learning in the future work.

**Questions:**

1. What is the computation cost of search a good scalarization among tasks? Is there any method to speed up the searching for the optimal loss weights when there are more than 2 tasks.

2. For the scalarization method, does the same group of hyperparameters works optimally for different scalarization of tasks?

**Limitations:**

Yes, they have addressed

**Strengths And Weaknesses:**

Strengths:

1. This paper summarizes the previous SOTA MTO methods and make a fair comparison between them and simple scalarization method on NLP translation task.

2. The comparison of SoTA MTO methods and simple scalarization is valuable since it shows with optimal scalarization, the static weighting of different tasks give competitive (or even better) performance than complex MTO methods.

3. Also the analysis of variation of MTL performance with different hyperparameters is very interesting and insightful.

Weaknesses:

1. Even though the optimal scalarization gives as good performance as other MTO methods, authors seem does not count in the computation cost to find the optimal weighting for different tasks, especially when there are more than 2 tasks.

2. Since the hyperparameter is very important to the MTL performance, it is unclear whether the same group of hyperparameters is optimal for all different loss weights among tasks.

---

> ### Author Response · Authors · 2022-08-02
> **Response to Reviewer i2dN**
>
> We thank the reviewer for their supportive feedback.
>
> We indeed agree with the reviewer that performing a grid search for the task weights (as we have done here) is computationally infeasible (especially for a large number of tasks). Note that, with the analysis that we have presented in this paper, we hope to encourage a shift in the thinking in the MTO community along the following lines:
> * Expensive MTO algorithms that promise significant reduction in task interference, simply find a solution in the scalarization solution space. Despite their computational costs, MTO solutions can be inferior to well-tuned scalarization solutions (Figure 5).
> * As such, a more fruitful avenue of research is to find efficient ways to search the scalarization solution space.
> In the text, we point out several papers that aim to perform such efficient exploration via bandits [4, 5]. We will augment the text to ensure the message is not lost.
>
> Regarding scalarization hyper-parameters, we have observed only minimal variation in the optimal hyper-parameters as we change the task weights. To highlight this observation, in Figure 16, we have plotted the change in the hyper-parameters of the Pareto optimal points for City Scapes dataset. We observe that the optimal hyper-parameters are all clustered in a small section of our search-space. For the final version of the manuscript, we will add additional hyper-parameter plots to cover the rest of the tasks studied in our analysis.
>
> [4] Graves et al. “Automated curriculum learning for neural networks” (2017)
>
> [5] Kreutzer et al. “Bandits don’t follow rules: Balancing multifacet machine translation with multi-armed bandits.” (2021)

---

### Official Review · Reviewer_s8zi · 2022-07-11

**Rating:** 6
**Confidence:** 3
**Soundness:** 4 excellent
**Presentation:** 2 fair
**Contribution:** 3 good

**Summary:**

This paper brings into question the efficacy of multi-task optimization methods, finding that when adequately tuning the baselines, MTO methods yield solutions similar to optimizing a weighted average of each tasks loss. They show that sensitivity to training hyper parameters lead to an often unfair comparison across methods and drastically different results on the same benchmarks, and provide solutions to standardize the evaluation of these methods.

**Questions:**

In 4.1 the authors mention that for the English to {French, Chinese} task they "anticipate a large degree of interference among the tasks"; what exactly does this mean?

In 4.1, I did not understand this baseline: "The blue dashed line corresponds to the Pareto front achieved via proportional sampling with English→French sampling rate ranging from 10% to 90%."Why do you only sample English to French if the task is English to French and Chinese? Am I misunderstanding the evaluation?

Is RLW an MTO algorithm or the baseline of taking the weighted average of the losses?

What dataset is used in 4.1?

Is test cross-entropy the usual evaluation metric used for NMT? What about things like BLEU score?

**Strengths And Weaknesses:**

The issue of insufficient hyperparameter tuning leading to the illusion of performance gains is incredibly important and often overlooked, and this paper does an excellent job of highlighting this fact. The claims of the paper were clearly laid out and the evaluation of sensitivity to hyperparameters was thorough.

However, I have two main concerns. The first is the organization of Section 4. I don't fully understand what the experimental setup is, the methods evaluated, and the results laid out specifically for the neural machine translation were not easy to follow. I realize that there is limited space so many experimental details can't be put in the main paper, but more summarization of the task setup and evaluation metrics would clear things up a lot. Also more of a suggestion than critique, but I wonder if employing a hyperparameter stability evaluation similar to [5] would be useful in illustrating these methods sensitivity to hyperparameters.

Second, the issue of hyper parameters not being swept correctly when evaluating methods has been heavily explored in the past few years [1-4], and the issue of reproducibility of results is also a widely accepted issue in the field. Ultimately I believe this paper provides novel contributions in its evaluation and specific focus on MTO methods, but these previous works should be mentioned in the related works.

With all of this in mind I believe that the ideas in this paper are of interest to the NeurlIPS community, although the presentation of the results needs clarification and the related works mentioned should be added. Overall a very interesting read :D

[1] Su et al. "A Comparison of Strategies for Source-Free Domain Adaptation"

[2] Musgrave et al. "Unsupervised Domain Adaptation: A Reality Check"

[3] Gulrajani et al. "In Search of Lost Domain Generalization"

[4] Greff et al. "LSTM: A Search Space Odyssey"

[5] Xiao et al. "Early Convolutions Help Transformers See Better"

---

> ### Author Response · Authors · 2022-08-02
> **Response to Reviewer s8zi**
>
> We thank the reviewer for their supportive comments. Following your suggestions, we will include a more substantial discussion of the evaluated methods to the appendix for the final version. Following the example of Radosavovic et al,  we have added additional figures to the appendix that examine the effect of hyper-parameter variability in the final results (Figures 15 & 16. Figure 10 was also previously included). Finally, we have added the missing citations to the manuscript.
>
> Response to questions raised in the review:
> * In 4.1 the authors mention that for the English to {French, Chinese} task they "anticipate a large degree of interference among the tasks"; what exactly does this mean? Prior literature has observed that when a model is trained on high-resource languages with minimal vocabulary overlap, the aggregate performance suffers (See [1], Section 4.1). This phenomenon has often been attributed to these different languages competing for the model capacity (i.e. task interference). French & Chinese certainly come from different linguistic / script families. As such, we anticipate task interference to be significant for these models.
>
> * In 4.1, I did not understand this baseline: "The blue dashed line corresponds to the Pareto front achieved via proportional sampling with English→French sampling rate ranging from 10% to 90%."Why do you only sample English to French if the task is English to French and Chinese? Am I misunderstanding the evaluation? In our setup (similar to most multilingual NMT applications) the training data is a mix of En->Fr and En->Zh sentences. Unfortunately, training data points of the form (En, Fr, Zh) are rather hard to gather. As such, we train our NMT models on a mix of En->Fr and En->Zh with different portions of each language.
>
> * Is RLW an MTO algorithm or the baseline of taking the weighted average of the losses? Random Loss Weighting (RLW) is an MTO algorithm proposed by Lin, Ye, Zhang “A Closer Look at Loss Weighting in Multi-Task Learning” (2022). In our paper, we examine two variations of this method (Normally distributed weights and Dirichlet distributed weights).
> What dataset is used in 4.1? All experiments of Section 4.1 use WMT datasets provided by the TFDS package. The details of these datasets are presented in Table 1.
>
> * Is test cross-entropy the usual evaluation metric used for NMT? What about things like BLEU score? Great question. The proper approach to evaluate NMT models is an active area of research with many metrics currently proposed in the literature [2, 3]. For the sake of uniformity (and to avoid the complexities of Beam-Search decoding), we present test xentropy in the main text and BLEU scores in the appendix (Figure 12). As the results suggest, the results of these experiments closely align.
>
>
> [1] Arivazhagan et al “Massively Multilingual Neural Machine Translation in the Wild: Findings and Challenges” (2019)
>
> [2] Freitag et al (2020) “BLEU might be guilty but references are not innocent.”
>
> [3] Freitag et al (2019) “ APE at Scale and Its Implications on MT Evaluation Biases.”

---

> > ### Comment · Reviewer_s8zi · 2022-08-07
> > **Thank you for the updates.**
> >
> > I appreciate your incorporation of my recommendations and clarifying my questions on the experimental setup. I believe that this paper will be of value to both the MTO community as well as the ML community as a whole.

---

### Meta-Review · Area_Chair_dhYD · 2022-08-26

**Recommendation:** Accept
**Confidence:** Certain

**Metareview:**

This paper reveals several important facts about multi-task optimization methods by providing a comprehensive benchmarking of MOT methods. They find that with carefully tuning the hyper-parameter,  MTO methods give similar results compared to simply averaging the weight of each task.  Then they go deeper into why this has not been noticed in previous work by studying the evaluation step and provide solutions to standardize the evaluation of these methods.
In particular, the reviewers have appreciated the comprehensive evaluation and tenable findings based on their experiments. Hence, I recommend that the paper be accepted.

**Award:**

No

---

### Decision · Program_Chairs · 2022-09-14

Accept